# Double Mutations in Succinate Dehydrogenase Are Involved in SDHI Resistance in *Corynespora cassiicola*

**DOI:** 10.3390/microorganisms10010132

**Published:** 2022-01-09

**Authors:** Bingxue Sun, Guangxue Zhu, Xuewen Xie, Ali Chai, Lei Li, Yanxia Shi, Baoju Li

**Affiliations:** Institute of Vegetables and Flowers, Chinese Academy of Agricultural Sciences, Beijing 100081, China; sunbingxuechina@163.com (B.S.); zhuguangxuer@163.cm (G.Z.); xiexuewen@caas.cn (X.X.); chaiali@caas.cn (A.C.); caulilei@163.com (L.L.)

**Keywords:** succinate dehydrogenase inhibitors, *Corynespora cassiicola*, double mutation, SDHI-resistance, fitness

## Abstract

**Simple Summary:**

With the application of fungicide in agriculture, resistance to fungicide has become a serious problem. It is important to assess the evolution of resistance for development of disease prevention and control. We confirmed, by site-directed mutagenesis, that single mutations conferring moderate or low resistance are more likely to evolve into double mutations conferring higher resistance under the selective pressure of SDHI. However, the double mutations suffer large of fitness penalty than single mutation. We recommend that the use of SDHI in agriculture should be appropriately reduced or that other types of fungicides should be used to control plant diseases, such as dicarboximide fungicides (DCFs), to avoid the emergence of very resistant plant pathogens.

**Abstract:**

With the further application of succinate dehydrogenase inhibitors (SDHI), the resistance caused by double mutations in target gene is gradually becoming a serious problem, leading to a decrease of control efficacy. It is important to assess the sensitivity and fitness of double mutations to SDHI in *Corynespora cassiicola* and analysis the evolution of double mutations. We confirmed, by site-directed mutagenesis, that all double mutations (B-I280V+D-D95E/D-G109V/D-H105R, B-H278R+D-D95E/D-G109V, B-H278Y+D-D95E/D-G109V) conferred resistance to all SDHI and exhibited the increased resistance to at least one fungicide than single point mutation. Analyses of fitness showed that all double mutations had lower fitness than the wild type; most of double mutations suffered more fitness penalties than the corresponding single mutants. We also further found that double mutations (B-I280V+D-D95E/D-G109V/D-H105R) containing low SDHI-resistant single point mutation (B-I280V) exhibited higher resistance to SDHI and low fitness penalty than double mutations (B-H278Y+D-D95E/D-G109V) containing high SDHI-resistant single mutations (B-H278Y). Therefore, we may infer that a single mutation conferring low resistance is more likely to evolve into a double mutation conferring higher resistance under the selective pressure of SDHI. Taken together, our results provide some important reference for resistance management.

## 1. Introduction

Corynespora leaf spot (CLS), caused by *Corynespora cassiicola*, has become one of the most important plant diseases. The plant pathogenic fungus *C. cassiicola* can infect more than 530 species of plants, including *Hevea brasiliensis*, cucumber, tomato, cowpea, and other vegetables, causing substantial economic losses [1,2,3]. Due to the lack of resistant cultivars, chemical fungicides are the main method to control *Corynespora* leaf spot. Many years ago, the control of *Corynespora* leaf spot was mainly dependent on benzimidazole fungicides, dicarboximide fungicides, and quinone outside inhibitors [4]. Due to the decline in the efficacy of these fungicides and the emergence of resistance [5,6], these fungicides are gradually being replaced.

Subsequently, a new generation of succinate dehydrogenase inhibitors (SDHI, e.g., fluopyram and boscalid) have been widely used in the field for disease control since 2010 (www.chinapesticide.org.cn/, accessed on 12 October 2020) because of its broad spectrum of fungicidal and bacteriostatic activity [7,8]. SDHI inhibit fungal respiration by binding to the ubiquinone-binding site (Q-site) of mitochondrial succinate dehydrogenase (SDH) complex II. Although SDH is a mitochondrial protein, the genes encoding SDH are the only family of mitochondrial complex genes located in the nucleus [7]. However, the incidence of SDHI resistance in *C. cassiicola* populations has gradually increased in recent years due to the large-scale reuse of these fungicides and their single mode of action, leading to the failure to control *Corynespora* leaf spot [5,9,10,11]. In recent studies, FRAC summarized the primary SDHI resistance mechanisms up to 2021 in 15 phytopathogenic fungi (www.frac.info, accessed on 12 October 2020). The primary SDHI resistance mechanisms are single point mutations in the SDHB/C/D genes of succinate dehydrogenase that result in amino acid substitutions [8,12,13]. Briefly, homologous substitutions of fifteen residues in the SDHB subunit, 13 residues in the SDHC subunit, and seven residues in the SDHD subunit were detected in resistant mutants; substitutions included B-278R/L/Y, C-N75S, and D-D95E. In *C. cassiicola*, nine genotypes that alter the amino acid sequence of the SDH complex were reported in previous studies, including B-H278R/Y, B-I280V C-S73P, C-N75S, D-S89P, D-D95E, D-H105R, and D-G109V [9,10,11]. Four genotypes (B-H278R/Y, C-S73P, and D-G109V) were both detected in China and Japan, while some genotypes were only found in China or Japan; e.g., B-I280V, C-N75S, D-D95E, and D-H105R genotypes were only found in China; C-S89P was only detected in Japan. Previous studies have also shown that various mutations in the SDH subunit exhibited different levels of resistance to SDHI and different fitness costs [14,15,16,17]. In some genotypes, cross-resistance to boscalid, penthiopyrad, and carboxin was also observed in various plant pathogens [5,11]. A lack of cross-resistance to boscalid and fluopyram was also found in two genotypes, B-H278Y/R, which led to the replacement of histidine by tyrosine or arginine in the SdhB subunit [5].

In recent years, some researchers have reported that strains with double point mutations in the same or different subunits in SDH can alter sensitivity to SHDIs. The current hypothesis for the emergence of double mutations is that double mutations in the target gene were derived from a single point mutation via intragenic recombination or secondary mutation under the high selective pressure of the fungicide [18,19]. We have been tracking double mutations affecting resistance to SDHI in plant pathogens (Appendix A), such as C-N86S+B-N225T, C-N86S+C-T79N, and C-N86S+C-L85P in *Mycosphaerella graminicola*, and B-H278Y/R and B-H277R/Y+C-H134R in *Alternaria solani* and *A. alternata* [20,21,22]. Most of the double mutations showed similar or higher resistance to fungicides than isolate with single mutations in the target gene [6]. However, the sensitivity to SDHI of double mutations has not been reported in *C. cassiicola.* With further fungicide applications in field, it is essential to clear the resistance risk to SDHI of double mutations in *C. cassiicola.*

In our present study, site-directed mutagenesis was performed on a single-point mutant isolates to generate isolates with double mutations in the *SDH* gene. The objectives of the present study were to study (I) the relationship between double mutations and SDHI resistance; (II) whether double mutations in the *SDH* gene increase or decrease SDHI resistance; (III) the fitness costs of these double mutations; and (IV) the evolution of double mutations.

## 2. Materials and Methods

### 2.1. Isolates and Fungicides

Isolates carrying a single point mutation in SDH were obtained from a previous study [23]. Double mutation isolates were derived from the single point mutant isolate. All the mutants shared the same genetic background; their parent isolate was SD1 (wild type, WT), which was sensitive to boscalid. All the isolates used in this study are listed in Appendix A.

Boscalid (97%) and carboxin (98%) were provided by Shanxi Meibang Pesticide Co., Ltd, Shanxi, China. Fluopyram (98%) was obtained from Junkai (Tianjin, China) Chemical Co. Tanjin, China. Penthiopyrad (98.8%) was provided by Mitsui Chemicals Agro., Inc. Tokyo, Japan.

### 2.2. Generation and Identification of Double Mutations

To obtain *C. cassiicola* transformants with double mutations in the succinate dehydrogenase subunits, site-directed mutagenesis was performed. In our research, the process of site-directed mutagenesis can be divided into three steps: first, the construction of a gene replacement vector; second, genetic transformation of *C. cassiicola*; and third, confirmation of double mutation.

First, the gene replacement vector was constructed by double-joint PCR, as previously described [23]. Three rounds of PCR were performed to generate the gene replacement vector. In the first round of PCR, upstream, downstream, and selection marker genes (Trpc-Neo) were amplified. In the subsequent PCR, these fragments were connected by double-joint PCR. In the last round of PCR, 1 µL of the connected product was used as the DNA template in nested PCR to amplify the replacement vector, and the vector was confirmed by sequencing (Biomed, Beijing, China).

Second, the genetic transformation of *C. cassiicola* was carried out as described previously [23] and was divided into 2 steps. First, protoplasts were prepared and collected. Then, l mL of conidial suspension from the edge of a 5-day-old colony was added to YEPD medium (20 g of glucose, 10 g of peptone, and 3 g of yeast extract in 1 L of deionized water). After, the YEPD medium was incubated for 36 h at 160 rpm and 28 °C. Subsequently, mycelia were harvested from the YEPD medium and washed twice with distilled water. Then, 0.1 g fresh mycelia were treated with 15 mg/mL driselase (Sigma-Aldrich, St. Louis, MO, USA) and digested at 80 rpm and 30 °C for 2.5 h. The enzyme solution was then filtered to eliminate mycelial residues. Protoplasts were isolated from the cell debris by filtration and then washed twice with STC (0.8 M sorbitol, 0.05 M Tris and 50 mM CaCl_2_). Protoplasts were resuspended in SPTC (STC with 40% *w*/*v* PEG6000). The next step was the transformation of the protoplasts. The concentration of protoplasts was adjusted to 2 × 10^7^~2 × 10^9^/mL. The solution for transformation containing 200 µL of protoplasts, 4–6 µg of gene replacement vector and 10 µL of heparin sodium was placed on ice for 1 h. Then, 1 mL of SPTC was added to the solution and incubated at room temperature for 1 h. The solution for transformation was mixed into 200 mL of RM medium (0.5 g of yeast extract, 0.5 g of casein hydrolysate, 0.7 M of sucrose, and 16 g of agar in 1 L of deionized water) and incubated at 28 °C. After 12 h, the RM plates were overlaid with SRM medium (0.5 g of yeast extract, 0.5 g of casein hydrolysate, 342.3 g of sucrose, and 10 g of agarose in 1 L of deionized water) with 100 µg/mL of G-418.

The final site-directed mutagenesis step was the confirmation of double mutations. The inserted position of the gene replacement vector, purity of the double transformants (*C. cassiicola* was heterokaryotic) and single-copy insertion of the cassette were verified. Transformants were picked after 5~7 days and cultured on PDA medium (200 g of potato, 20 g of glucose, and 15 g of agar) containing 100 µg/mL of G-418. DNA was extracted from transformants by the CTAB method. Primer pairs P3/P4 and P5/P6 (Appendix A) were used to confirm that the gene replacement vector was inserted into the correct site in *SDHB* gene. The primer pair P7/P8 was used to confirm the purity of the double transformants. Southern blot analysis was performed by digesting genomic DNA with EcoRV and labeling a probe using the Dig High Primer DNA Labeling and Detection Starter Kit I (Roche, Shanghai, China). The presence of each double mutation in the transformants was demonstrated by sequencing the entire *SDHB/D* gene.

### 2.3. Activity Analyses of Succinate Dehydrogenase

The mitochondria of each strain were extracted using the Fungi Mitochondrial Extraction Kit (Bestbio, Shanghai, China). Five mycelial plugs taken from the periphery of a 5-day-old colony were added to YEPD medium and incubated at 160 rpm and 28 °C for 36 h. The mycelium was collected using three layers of gauze and washed twice with PBS [24]. All subsequent procedures were performed on ice. Fresh mycelia were disrupted with liquid nitrogen and resuspended in solution A. The suspension was centrifuged sequentially at 500, 1000, and 2000× *g* for 5 min and 11,000× *g* for 20 min. After each centrifugation, the suspension was transferred to a new tube. The acquired pellet was resuspended in 400-µL solution B and then centrifuged again at 11,000× *g* for 15 min. The mitochondria were resuspended for preservation in solution C. The protein concentration of the mitochondria was determined according to the Bradford protein assay. The mitochondrial fraction obtained was used for the activity analyses of succinate dehydrogenase. The activity of succinate dehydrogenase was assessed using DCPIP (2,6-dichlorophenol-indophenol) as a terminal electron acceptor. All subsequent procedures were performed according to the succinate dehydrogenase (SDH) assay kit (Nanjing Jiancheng, Nanjing, China). DCPIP reduction was recorded at 600 nm, after 5 and 65 s with a spectrophotometer. Enzyme activity was calculated as the reduction in absorbance per minute per mg of protein.

### 2.4. The Expression of the SDHA/B/C/D Genes

According to a previous study, the expression of the *SDH* genes in the double mutations was analyzed [23]. Briefly, five mycelial plugs were incubated in YEPD medium and incubated at 160 rpm and 28 °C for 2 days. Total RNA was extracted with a Plant Genomic RNA Kit (Tiangen, Beijing, China). RNA was used to synthesize cDNA with the FastKing RT Kit (Tiangen, Beijing, China). The expression of the *SDHA/B/C/D* gene was determined by RT-qPCR in an ABI 7500 real-time detection system (Thermo Fisher Scientific, MA, USA). The primers used for the expression of the *SDHA/B/C/D* genes are listed in Appendix A. The relative expression levels were determined according to the 2^−ΔΔC^ (t) method. All data were normalized to the elongation factor 1-α (*EF1-α*) gene.

### 2.5. Fungicide Sensitivities Test

The sensitivity of the single- and double-mutation isolates to SDHI fungicides (boscalid, carboxin, fluopyram, and penthiopyrad) was assessed by mycelial growth methods as previously described [23]. Briefly, the fungicide (boscalid, carboxin, fluopyram, penthiopyrad) concentrations used for single and double mutations were 0, 0.01, 0.1, 0.5, 1, 10, and 30 µg/mL. Mycelial plugs (6 mm in diameter) taken from a 5-day-old colony were added to YBA (yeast extract 1%, Bacto Peptone 1%, sodium acetate 2%, agar 1.5%) plates (containing different concentrations of fungicides). The plates were then incubated at 28 °C for 5 days. Finally, the colony diameters were measured, and the EC_50_ values were calculated.

### 2.6. Comparison of Resistance to SDHI in Single and Double Mutations

According to Wadley’s method, we calculated the synergistic coefficient to compare the resistance to SDHI in the single and double mutations [25]. The formula is as follows:Synergistic coefficient (SC) = (Actual EC_50_ (AE) of double mutation (A + B))/(Theoretical EC_50_ (TE) of double mutation (A + B))(1)
AE = (Actual EC_50_ (AE) of single point mutant)/(EC_50_ of standard single point mutant)(2)
TE of double mutation (A + B) = AE(A) × 0.5 + AE(B) × 0.5(3)
where A and B represent the single point mutation and A + B represents the double mutation. When the SC value is less than 0.8, it indicates an antagonistic effect; when the SC value is between 0.8 and 1.2, it indicates an additive effect; and when the SC value is greater than 1.2, it indicates that there is a synergistic effect.

### 2.7. Fitness Studies on Mutants

To assess the ability of a plant–pathogenic fungus to survive in the environment, some components of fitness were measured, including mycelial growth rate, production and germination of conidia, pathogenicity, and sensitivity to various environmental stresses. To analyze mycelial growth, mycelial plugs taken from the edge of a 5-day-old colony of the mutants were placed in PDA medium. The PDA medium was measured at 28 °C and then incubated in the dark at different temperatures (15, 20, 28, 32, 37 °C) for 10 days. Subsequently, the colony diameters and mycelial growth rates were measured.

For assessment of conidia production, mycelial plugs (6 mm diameters) taken from the edge of 10-day-old plates were placed in 90-mm Petri dishes containing PDA medium and incubated at 28 °C for 10 days in darkness. Subsequently, conidial suspensions were eluted and filtered by three layers of gauze. The conidial concentrations were quantified using a hemocytometer. Each isolate was cultured on three plates, and the experiment was performed two times.

To analyze conidial germination, the concentration of the conidial suspension was adjusted to 10^4^/mL, and 30 µL of each conidial suspension was placed on a slide and incubated at 28 °C for 4–6 h. Following this incubation, the number of germinated spores among 100 randomly selected spores was assessed under a microscope. Each isolate or mutant was cultured on three slides, and the experiment was performed three times.

To assess the pathogenicity of single or double mutations, mycelial plugs were taken from the margin of a 5-day-old colony of each isolate and mutant. The plugs were then placed on a cucumber leaf, each of which was of the same foliar age and similar size. Then, these leaves were incubated at 28 °C and 80–100% RH for 5 days. The pathogenicity was then evaluated by measuring the diameter of the surrounding lesion. The experiment was performed three times.

To test susceptibility to various environmental stresses (osmotic stress, metal ions, oxidative and salicylhydroxamic acid stress, and cell wall damage), mycelial plugs (6 mm in diameter) obtained from the periphery of 5-day-old colony of each isolate were incubated with several compounds (1 M of glucose, 1 M of NaCl, 1.2 M of KCl, 0.3 mg/mL of Congo red, 0.2 M of CaCl_2_, and 0.1 mg mL^−1^ of sodium dodecyl sulfate (SDS)). For each set of test conditions, we inoculated three Petri dishes in the dark. After 9 days of incubation at 28 °C, the colony diameter of each isolate was measured. The percentage inhibition of mycelial radial growth (PIMG) was calculated using the following equation: PIMG = [(C-N)/(C-6)] × 100, where C is the colony diameter of the untreated control and N is the colony diameter after treatment.

## 3. Results

### 3.1. Confirmation of the Double Mutations

To obtain *C. cassiicola* transformants with double mutations, site-directed mutagenesis was performed. First, the 4484-bp gene replacement vector was constructed by double-joint PCR, containing the left and right homologous arms (gray fragments in Figure 1B, 1046 and 1167 bp) of the *SDHB* gene and the mutational *SDHB* gene *(SDHB* + Trpc + neo*, 2271 bp, middle part in Figure 1B). Subsequently, the non-mutational *SDHB* gene (Figure 1A) was replaced with the gene replacement vector (Figure 1B) to generate double mutation transformants (Figure 1C). Finally, the transformants with double mutations were confirmed by following three aspects: the inserted position of the gene replacement vector, the purity of the double transformants (*C. cassiicola* was heterokaryotic), and the single-copy insertion of the cassette.

Transformants with double mutations connected into the locus yielded a single 2575- or 2791-bp band with primers P3/P4 or P5/P6, respectively, whereas those in which integration was ‘random’ yielded no band. These primers did not amplify any fragment from the wild-type (WT) isolate (Figure 1D(Up),(Down)). The purity of the transformants was confirmed by primers P7/P8 or P9/P10. Homokaryotic transformants with double mutations yielded a single 3256- or 2192-bp band with primers P7/P8 and P9/P10, respectively, whereas heterokaryotic transformants with double mutations yielded double bands of 3256 and 2091 bp by P7/P8 primers and 2192 and 1027 bp by P9/P10 primers. Primers P7/P8 or P9/P10 amplified 2091 and 1027 bp from the wild-type isolate (Figure 1D(Up),(Down)).

We also confirmed the single-copy insertion of the cassette by Southern blotting genomic DNA digested with EcoRV. When probed with the partial *SDHB* gene region, the genomic DNA of the double mutation isolates digested with EcoRV had a single 4551-bp hybridized DNA fragment instead of the 3386-bp fragment found in the parental isolate (Figure 1F). Finally, the presence of each mutation was demonstrated by sequencing the entire *SDHB* gene.

### 3.2. Sensitivity to SDHI Fungicides

#### 3.2.1. Sensitivity of Double Mutation to SDHI

Sensitivity to SDHI fungicides (carboxin, boscalid, fluopyram, and penthiopyrad) was tested for all double mutations and parental isolates. All the data was shown in Table 1. All double mutations were resistant to boscalid. However, the EC_50_ values varied among the double mutations. The double mutations B-I280V+D-D95E, B-H278R+D-D95E, and B-H278Y+D-G109V were classified as moderate resistance (MR), with EC_50_ values ranging from 2.472~4.062 µg/mL. However, three double mutations, including the B-H278Y+D-D95E, B-I280V+D-G109V, and D-B-H278R+D-G109V genotypes, were identical to high resistance (HR, EC_50_ values: 13.863~16.822 µg/mL). The EC_50_ values of B-I280V+D-H105R were >25 µg/mL and were identified as conferring very high resistance (VHR).

We also assessed these mutants for their sensitivity to fluopyram. All of the mutants were resistant to fluopyram, but the resistance level varied among the double mutations, with RF values ranging from 1.357 to 130.30. In isolates carrying only B-H278Y/R mutations, negative cross-resistance was observed between boscalid and fluopyram, whereas no negative cross-resistance was found in the double mutations carrying the B-H278Y/R single point mutation, with RF values ranging from 1.357 to 39.29 for fluopyram and from 16.48 to 113.40 for boscalid. Further evaluation revealed that most double mutations showed either low to moderate resistance to carboxin (5 < RFs < 25) or moderate to high resistance to carboxin (19 < RFs < 216). Only one double mutation was classified as conferring low resistance to penthiopyrad, with an RF value of 3.86.

#### 3.2.2. Comparison of Resistance to SDHI in Single or Double Mutations

To further analyze the effect of double mutations on SDHI resistance compared with single mutations, synergistic coefficients (SCs) were calculated. However, the effect of double mutations varied among the different SDHI and mutants. All of the double mutations showed that the increased resistance to at least one fungicide (Table 1 and Table 2, Figure 2). The effect of B-I280V+D-D95E and B-I280V+D-H105R double mutations were synergistic in all SDHI, with SC values ranging from 1.43 to 11.40. In the B-I280V+D-G109V and B-H278R+D-G109V double mutations, synergistic effects in carboxin, boscalid, and penthiopyrad were also found, but antagonistic to fluopyram (SC = 0.74 and 0.61, respectively). Three mutations (B-H278Y+D-D95E, B-H278R+D-D95E, and B-H278Y+D-G109V) was mainly manifested in additive or antagonistic effects on most SDHI (Table 1, Figure 2).

#### 3.2.3. Cross-Resistance Analysis

The cross-resistance among SDHI was analyzed based on the lg (EC_50_) values. Regression analyses and Spearman’s correlation tests revealed no significant cross-resistance between fluopyram and boscalid (r_s_ = 0.551, *p* = 0.157), fluopyram and carboxin (r_s_ = 0.263, *p* = 0.528), or carboxin and boscalid (r_s_ = 0.699, *p* = 0.054) (Figure 3), while strong correlations were observed between penthiopyrad and all the SDHI tested (carboxin: r_s_ = 0.713, *p* = 0.040; boscalid: r_s_ = 0.838. *p* = 0.009; fluopyram: r_s_ = 0.786, *p* = 0.021) (Figure 3).

### 3.3. Fitness Analysis of Double Mutation Isolates

The fitness of fungi, including the production and germination of conidia, pathogenicity, mycelial growth, and the sensitivity to various environmental stresses, plays an essential role in the ability of the fungus to spread and colonize in new environments [16].

In the evaluation of pathogenicity, we found that the B-H278R+D-D95E and H-H278Y+D-D9E double mutations produced significantly smaller lesions than the wild type, but were not found in the other double mutations (Figure 4E). In terms of the spore germination rate, the double mutation B-I280V+D-G109V and B-H278R+D-G109V showed a lower spore germination rate than the WT isolate (76.40% of WT and 68.34%, respectively). However, other double mutations exhibited insignificant differences in germination rate of spore (Figure 4B). The production of conidia was significantly decreased in the B-H278R+D-G109V, B-H278Y+D-D95E, and B-H278Y+D-G109V mutants (5.5%, 19.6%, and 20.86% of WT, respectively) when compared to that in the wild type. However, the B-I280V+D-D95E mutant showed an increase in the production of conidia (Figure 4A). The growth rate of double mutation B-H278R+D-G109V was lower than wild type at 20–32 °C (18.75–52.5% of WT). Similar results were also found in other double mutations (B-H278Y+D-G109V, B-H278Y+D-D95E and B-I280V+D-H105R mutants) at different temperatures (Figure 4C).

When assessed for sensitivity to various environmental stresses, we found that the B-H278Y+D-D95E mutant exhibited a lower ability to cope with environmental stress, as evidenced by inhibition of osmotic stress, cell wall damage, and alternative respiration. A similar phenomenon was found in the B-I280V+D-D95E and B-I280V+D-G109V mutants, which showed increased osmotic stress sensitivity. However, three double mutations (B-I280V+D-G109V, B-H278R+D-G109V, and H278Y+D-D95E) showed significantly reduced sensitivity to paraquat (Figure 4D).

To accurately analyze the fitness of the double mutations, we scored their fitness. All double mutations had lower fitness scores than the wild type, and most double mutations suffered more fitness penalties than the corresponding single mutants (−18.75–68.75). Interestingly, the B-I280V+D-D95E double mutation showed higher fitness scores than the single mutant, but lower than the wild type (Appendix A).

### 3.4. SDHA/B/C/D Gene Expression in the Double Mutation

To determine whether the double mutations in the SDH subunit impact the four *SDH* gene expression levels, the expression of these genes in double mutations was quantified. Of the four *SDH* genes, only the expression of *SDHB* significantly differed in the double mutations compared to that of the wild type. The expression of the *SDHB* gene was 28.17-, 43.13-, and 22.96-fold higher in the double mutations B-H278R+D-D95E, B-H278R+D-G109V, and B-I280V+D-G109V, respectively, than that of the wild type. In contrast, the expression of the *SDHB* gene in the double mutations B-H278Y+D-D95E, B-H278Y+D-G109V, and B-I280V+D-H105R was not significantly different from that of the wild-type strain (Figure 5).

### 3.5. The Activity of SDH Enzyme

SDH activity was tested in the double mutations that contained two amino acid substitutions. Activity was determined in a 2,6-dichloroindophenol (DCPIP)-reduction assay in the presence of SDHI. Our results showed that SDH activity was reduced to some extent in all of the double mutations except for the isolate carrying the B-I280V+D-D95E double mutation. The SDH enzyme activity of wild type was 18.90 U/mg protein. Of all the double mutations tested, the B-I280V+D-D95E mutant showed the highest SDH activity, with 20.09 U/mg of protein. The other six double mutations (B-I280V+D-H105R, B-I280V+D-G109V, B-H278R+D-D95E, B-H278R+D-G109V, B-H278Y+D-D95E, and B-H278Y+D-G109V) had significantly lower SDH enzymatic activity than the wild-type isolate, and these six double mutations also exhibited lower activity than the corresponding single mutant (Figure 6).

Meanwhile, the correlation between SDH activity and sensitivity to SDHI were analyzed. Pearson’s correlation tests revealed strong negative correlations between sensitivity to penthiopyrad and SDH enzyme activity (r = −0.793, *p* = 0.033) (Table 3). However, no significant correlations between SDH enzyme activity and sensitivity to other SDHI were found.

## 4. Discussion and Conclusions

SDHI-resistance has become a serious problem in agriculture, leading to a decrease of control efficacy [20,26,27]. Several studies have shown that double mutation in *SDH* gene further reduces susceptibility to SDHI than single point mutation, leading to a reduction of control efficacy of SDHI fungicides [6,20,21]. However, double amino acid substitutions, resulting in SDHI-resistance, were not reported in *C. cassiicola*. In this research, site-directed mutagenesis was performed to create double mutations in the *SDH* gene, and these double mutations were analyzed for their resistance to SDHI and fitness, and to further understand the evolution of resistance.

Previous studies have shown that multiplex amino acid substitution in target genes of pathogens such as *SDH*, *CYP51,* and *β-tubulin* genes significantly decreased the sensitivity to corresponding fungicides compared with the single mutation [6,21,22,28]. Similar results were also observed in our study. All of the double mutations showed the increased resistance to at least one fungicide (Table 1 and Table 2, Figure 5), which may be explained by the change of spatial structure of the quinone-binding site [29,30]. This result indicated that the further decline in the efficacy to control CLS if double mutations in *SDH* gene occur in field by a combination of pre-existing single point mutations, interestingly, increased sensitivity to fluopyram and penthiopyrad (or antagonistic effect of double mutations on SDHI-resistance, SC < 0.8) were found in double mutations (B-H278Y+D-D95E/G109V and B-H278R+D-D95E). These results are in agreement with a previous study that showed that the B-H277Y+C-H134R double mutation in *Alternaria alternata* improved susceptibility to fluopyram and penthiopyrad compared to the single amino acid substitutions C-H134R and B-H277Y [20,22]. The double mutations in both studies contained the B-H278Y mutation. A possible hypothesis may explain this phenomenon. Modifications in the quinone-binding pocket refer to structural changes that modulate the level of the SDH/SDHI interactions, including the breakage of hydrogen bonds and hydrophobic centers [29] and changes in the topology of ubiquinone-binding sites [7,29,30].

A previous study reported that fungicide applications are major determinants of the change of resistance genotype in *Zymoseptoria tritici* [31]. This result was similar to our previous study, where the frequency of B-H278Y (VHR to boscalid and sensitivity to fluopyram) in the field significantly decreased from 2014–2019 under the application of fluopyram [11,23]. In our research, three double mutations (B-I280V+D-D95E, B-I280V+D-H105R, and B-I280V+D-G109V) in Group I had a synergistic effect (or enhancement of SDHI-resistance) with most SDHI fungicides. However, double mutations in Group III had an additive effect or antagonistic effect on most SDHI and included B-H278Y+D-D95E and B-H278Y+D-G109V. This result indicated that the double mutations in Group I, caused by combination of pre-existing single point mutations, may occur in field, failing to control CLS, because of increasing dosages and service times.

Interestingly, the synergistic coefficients of all the double mutations for carboxin were approximately 1.2, indicating that the structural changes in the ubiquinone domain caused by double mutations in the SDH enzyme had little effect on carboxin resistance. In contrast, changes in the structure of the ubiquinone domain had a greater impact on the resistance to boscalid, fluopyram and penthiopyrad. A possible hypothesis to explain this phenomenon is that the affinity between carboxin and the SDH enzyme is lower than that of the other SDHI, such as boscalid and fluopyram, as evidenced by the significantly lower activity (EC_50_ values) of carboxin compared to the other SDHI [9,11,27].

Negative or lack of cross-resistance relationships between boscalid and fluopyram have been found for the homologous substitution B-H278R/Y in plant pathogens [5,11]. However, no negative cross-resistance or relationship between boscalid and fluopyram was found in any of the double mutations carrying the B-H278R/Y substitution in our study, including the B-H278R/Y+D-D95E and B-H278R/Y+D-H105R substitutions (Table 1). Positive cross-resistance relationships between penthiopyrad and boscalid, carboxin, or fluopyram, respectively, were also found in double mutation isolates (Figure 4), suggesting that a further decrease in efficacy and an increase in the resistance of double mutations occur at increasing doses and dosing times.

Fitness was defined as the ability of a plant-pathogenic fungus to survive or spread in field [15,17,32,33]. In our study, all double mutations except B-I280V+D-D95E in the *SDH* gene showed lower fitness than single point mutation (Appendix A, Figure 4). The result was similar to that observed in the *Mycobacterium smegmatis* and *Alternaria solani* [20,34]. In *A. solani*, the growth rate of B-H278Y/R+C-H134R double mutation was lower than B-H278R/Y [20]. However, the fitness of the double mutations varies. Briefly, the double mutation in Group I had a lower fitness penalty than the double mutations of Group III, mainly in terms of the production of conidia and mycelial growth. Interestingly, double mutations in Group I carried B-I280V mutation with a low fitness cost, while double mutations in Group III contained the B-H278Y mutation with a high fitness cost. Therefore, we suggest that it is easier to mutate from low fitness penalty isolates.

Some factors, including fitness cost, resistance levels conferred by mutations, the application of fungicides, and populations, may affect the evolution of resistance in the field [15,16,35,36]. In our research, three double mutations (B-I280V+D-D95E, B-I280V+D-H105R, and B-I280V+D-G109V) in Group I showed increased resistance to most SDHI with lower fitness penalty, and the three double mutations contained the B-I280V mutation (low or moderate resistance to SDHI). However, the double mutations (B-H278Y+D-D95E and B-H278Y+D-G109V) in Group III had minimally additive or antagonistic effects on SDHI (similar or decreased resistance to SDHI), both carrying the B-H278Y mutation (very high resistance to boscalid) [11]. These double mutations suffer huge fitness costs in our study (Appendix A, Figure 4). Therefore, we speculate that single mutations conferring moderate or low resistance to SDHI with lower fitness penalty are more likely to evolve into double mutations conferring higher resistance under the selective pressure of SDHI. This phenomenon was also found in benzimidazole resistance and boscalid resistance [6,18]. Certainly, the genetic background of the population plays a necessary role in the evolution of fungicide-resistant [33]. Why would single point mutations with high resistance to SDHI not evolve into double mutations? Two possible explanations could account for this. First, high fungicide resistance levels are often associated with high fitness costs [37,38]. Second, populations with low to moderate resistance were more susceptible to SDHI selective pressure compared to populations with high levels of resistance, thus making mutations more likely to occur.

The SDH enzyme is a functional protein and plays an essential role in the tricarboxylic acid cycle [7]. In our research, various degrees of decline in SDH activity were found in all of the double mutations except the double mutation strain B-I280V+D-D95E. A similar result was observed for *SDHB* mutations in *Botrytis cinerea*, which subsequently affected respiration rates in mitochondria [24]. A strong correlation (r = −0.704, *p* = 0.007 Appendix A) between *SDHA* gene expression and SDH activity was found for the *SDH* mutations, which may be explained by the function that *SdhA* subunit was the primary site of succinate oxidation in the succinate-binding pocket [8]. In addition, massive overexpression (more than 40-fold of WT) of the *SDHB* gene was found, but not in *SDHA/C/D* gene (4–5-fold of WT) (Figure 5). A similar result was found in *Fusarium asiaticum*. Of the eight *FaSdh* subunits, only *FaSdhC1* expression dramatically increased by more than 100-fold compared with the parent strain [39].

In summary, we first reported the sensitivity to SDHI of double mutations in *C. cassiicola*. All double mutations exhibited that the increased resistance to at least one fungicide and fitness penalty than single amino acid substitution in succinate dehydrogenase. Furthermore, single mutations conferring moderate or low resistance are more likely to evolve into double mutations conferring higher resistance under the selective pressure of SDHI. Therefore, we should reduce the application of SDHI used in controlling leaf spot in greenhouse-cultivated cucumber, and we recommend using other types of fungicides for controlling this disease, such as dicarboximide fungicides (DCFs).

## Figures and Tables

**Figure 1 microorganisms-10-00132-f001:**
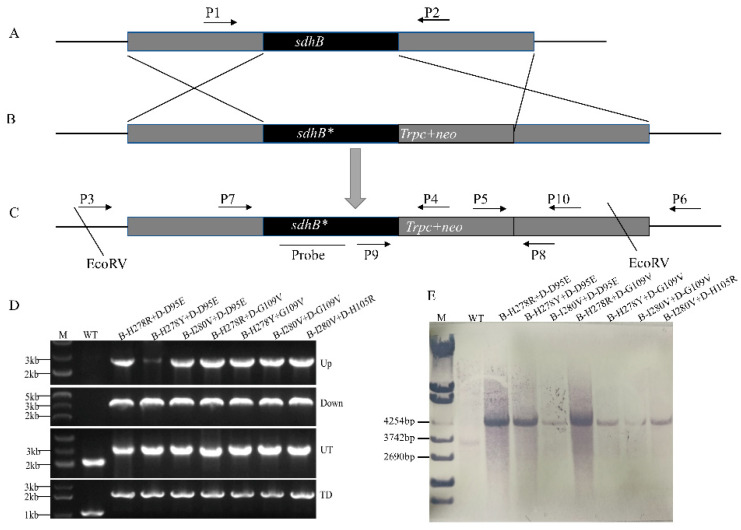
Generation and identification of *C. cassiicola SDHB* gene replacement mutants. (**A**–**C**) Schematic representation of the gene replacement strategy. The double mutations were obtained by gene substitution of a single point mutation (resistant mutants caused by a single point mutation in the *SDHD* gene, including D-D95E, D-H105R, and D-G109V) as the parent strain. (**A**) The *SDHB* target of the resistant mutants containing a point mutation in the *SDHD* gene. (**B**) The gene replacement vector was constructed by double-joint PCR. The two gray fragments represent the left and right homologous arms of the SDHB gene. The black fragment represents the target *SDHB* gene. The middle part represents the gene replacement cassette (*SDHB** + *Trpc* + *neo*) connecting the left and right homologous arms of *SDHB*. (**C**) Mutants carrying the double mutation. (**D**) PCR assay to verify the integration of the replacement cassette at the left connection (**Up**), and right connection (**Down**) points. PCR was also performed to indicate homozygosity of the replacement cassette at the left connection (**UT**) and right connection (**TD**) points. (E) Southern blot hybridization analysis of WT transformants using an 858-bp fragment from the *SDHB* gene as a probe. Genomic DNA was digested with EcoRV at the *SDHB* gene.

**Figure 2 microorganisms-10-00132-f002:**
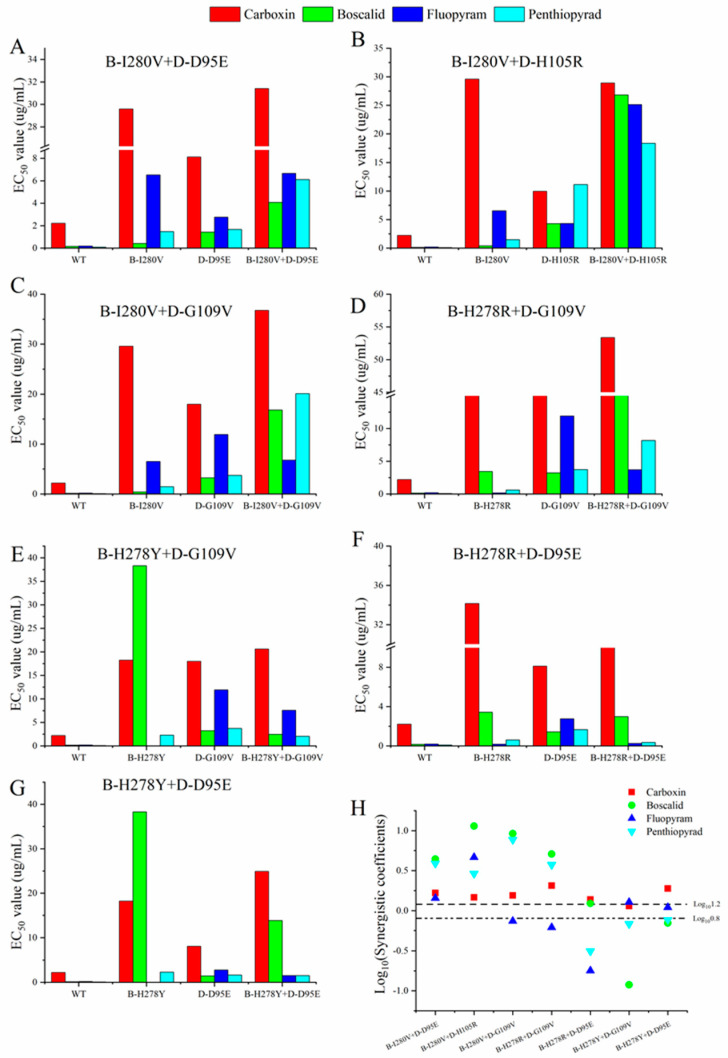
Comparison of SDHI resistance among the *C. cassiicola* wild-type and single or double mutation. (**A**–**G**) comparison of EC_50_ values of single and double mutation sensitivities. (**H**) comparison of synergistic coefficients of the double mutations. The synergistic coefficients were expressed on a Log_10_ scale.

**Figure 3 microorganisms-10-00132-f003:**
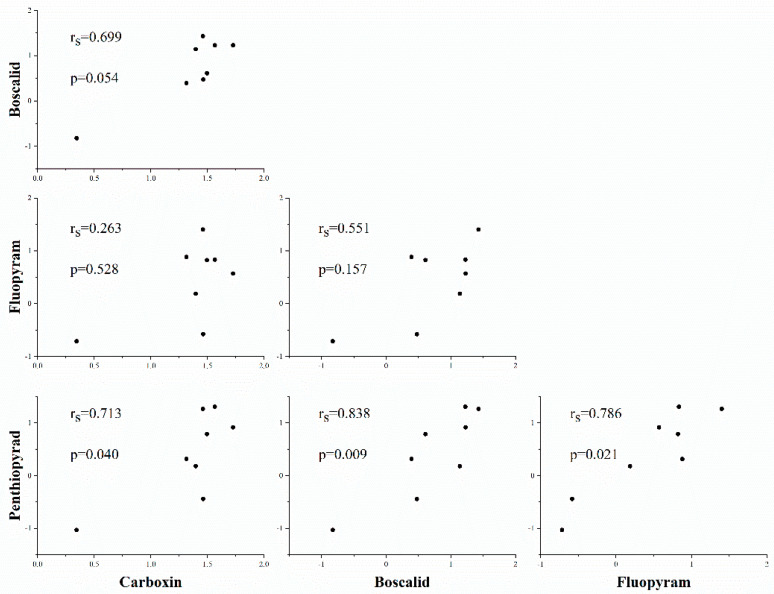
Spearman’s correlations between the resistance to three SDHI for *C. cassiicola* strains with various double mutations. Sensitivity data measured as EC_50_ (µg/mL) values are expressed on a log_10_ scale. *p* < 0.05 indicates that the correlation was statistically significant.

**Figure 4 microorganisms-10-00132-f004:**
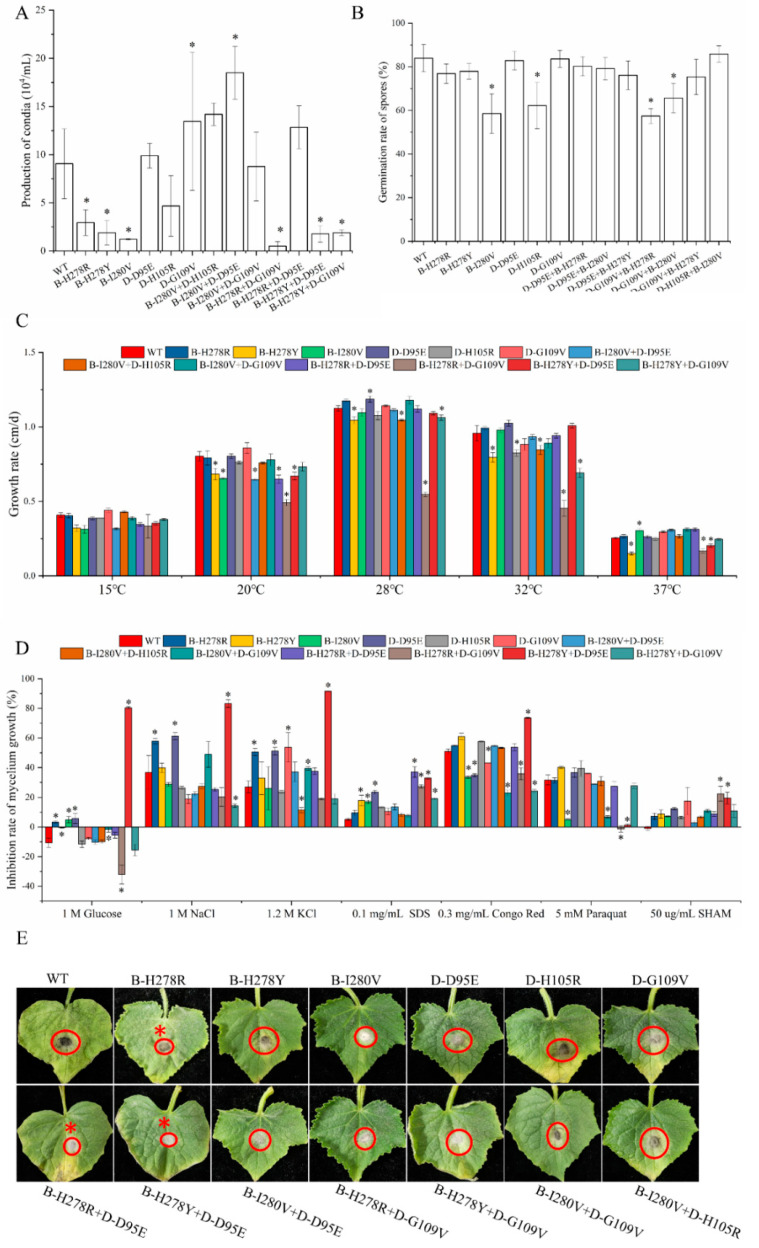
Analysis of fitness components among the double and single mutation isolate. * Indicates significant differences compared with WT (*p* < 0.05). (**A**) Conidial production of the double and single mutation isolate. Strains were cultured on PDA medium under continuous dark conditions for 10 days. (**B**) Germination rates of the double and single mutation strain after 4–6 h of incubation in 0.1% Tween 20 medium at 28 °C. (**C**) Mycelial growth rate of *C. corynespora* mutants at different temperatures. Mycelial growth was measured at 28 °C on 90-mm-diameter Petri dishes containing 20 mL of PDA medium. (**D**) Environment stresses include osmotic (generated by 1 M of glucose,1 M of NaCl, and 1.2 M of KCl), oxidative stress (generated by 5 mM of paraquat), cell wall damage (generated by 0.03 mg mL^−1^ of Congo Red and 0.1 mg mL^−1^ of SDS) and salicylhydroxamic acid (SHAM; generated by 50 μg mL^−1^ of SHAM). After 9 days of incubation at 28 °C, the percentage inhibition of mycelial radial growth (PIMG) was calculated. (**E**) Pathogenicity of the double and single mutation strains in the absence of fungicide treatment. Pathogenicity was analyzed after 5 days of incubation at 28 °C.

**Figure 5 microorganisms-10-00132-f005:**
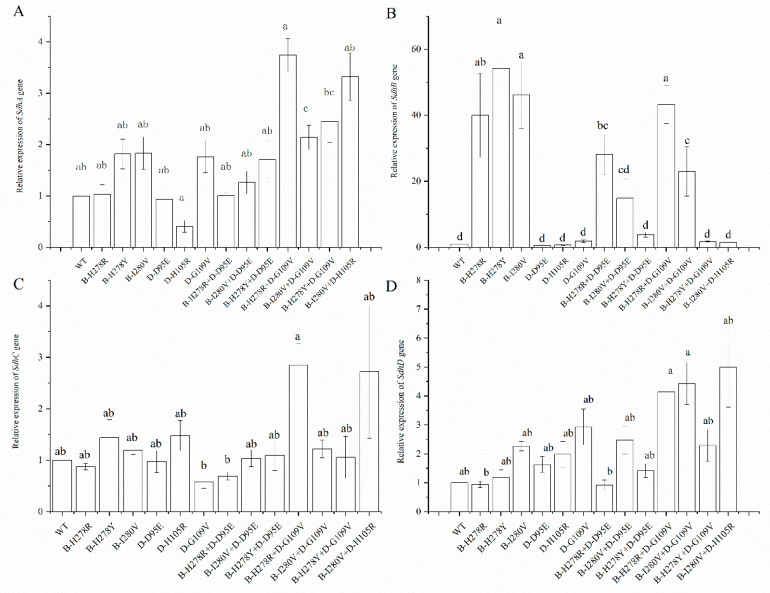
Relative expression levels of the *SDHA/B/C/D* genes in the double mutation. All data were normalized to *EF-1* gene expression, and relative changes in expression were analyzed by the ABI 7500 software. Different letters indicate significant differences (*p* < 0.05) (**A**) Expression of *SDHA* gene. (**B**) Expression of *SDHB* gene. (**C**) Expression of *SDHC* gene. (**D**) Expression of *SDHD* gene.

**Figure 6 microorganisms-10-00132-f006:**
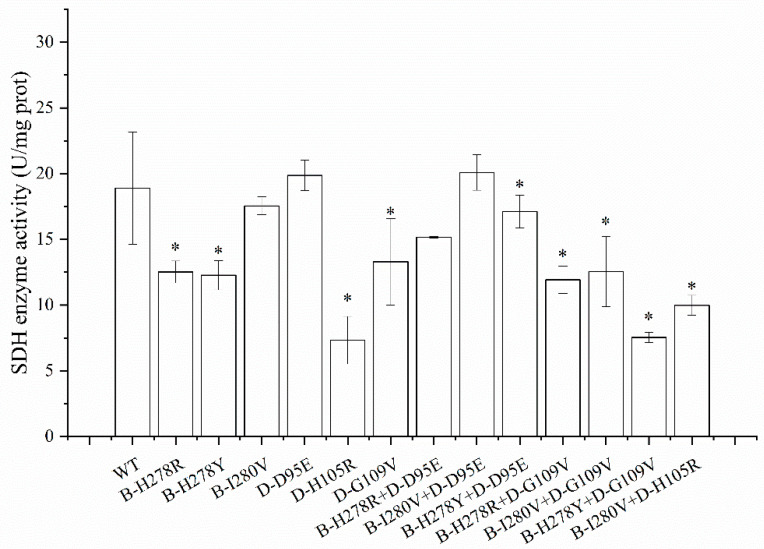
SDH activity of single or and double mutation. * indicates significant differences compared with WT (*p* < 0.05).

**Table 1 microorganisms-10-00132-t001:** Sensitivity of *C. cassiicola* to succinate dehydrogenase inhibitor (SDHI) fungicides.

Strain or Mutants	EC_50_ (µg/mL)			
	Carboxin	Boscalid	Fluopyram	Penthiopyrad
WT	2.212 ^a^ (1 ^b^)	0.15 (1)	0.193 (1)	0.093 (1)
B-I280V	29.584 (13.37)	0.413 (2.753)	6.534 (33.85)	1.469 (15.79)
B-H278R	34.143 (15.43)	3.435 (22.9)	0.168 (0.870)	0.615 (6.612)
B-H278Y	18.229 (8.240)	38.315 (255.4)	0.029 (0.150)	2.284 (24.55)
D-D95E	8.132 (3.676)	1.425 (9.5)	2.765 (14.32)	1.665 (17.90)
D-H105R	9.943 (4.495)	4.293 (28.62)	4.317 (22.36)	11.133 (119.7)
D-G109V	17.99 (8.132)	3.245 (21.63)	11.916 (61.74)	3.733 (40.13)
B-I280V+D-D95E	31.416 (14.20)	4.062 (27.08)	6.659 (34.50)	6.115 (65.75)
B-H278R+D-D95E	29.043 (13.12)	2.993 (19.95)	0.262 (1.357)	0.359 (3.860)
B-H278Y+D-D95E	24.924 (5.521)	13.863 (43.73)	1.534 (13.69)	1.508 (19.33)
B-I280V+D-G109V	36.758 (16.61)	16.822 (112.1)	6.799 (35.22)	20.101 (216.1)
B-H278R+D-G109V	53.372 (24.12)	17.023 (113.4)	3.71 (19.22)	8.172 (87.87)
B-H278Y+D-G109V	20.622 (9.322)	2.472 (16.48)	7.584 (39.29)	2.06 (22.15)
B-I280V+D-H105R	28.909 (13.06)	26.827 (178.8)	25.158 (130.3)	18.345 (197.2)

^a^ Values are indicated as represent the EC_50_. ^b^ Resistance factor.

**Table 2 microorganisms-10-00132-t002:** The synergistic activity of the combination of different single point mutations in SDHI resistance.

Double Mutation	Group	Carboxin	Boscalid	Fluopyram	Penthiopyrad
B-I280V+D-D95E	I	1.67 ^a^	4.42	1.43	3.90
B-I280V+D-G109V	I	1.55	9.20	0.74	7.73
B-I280V+D-H105R	I	1.46	11.40	4.64	2.91
B-H278R+D-D95E	II	1.37	1.23	0.18	0.31
B-H278R+D-G109V	II	2.05	5.10	0.61	3.76
B-H278Y+D-G109V	III	1.14	0.12	1.27	0.68
B-H278Y+D-D95E	III	1.89	0.70	1.10	0.76

^a^ Synergistic coefficients (SC) were calculated by the following formula: Actual EC_50_ (AE) of a single point mutation (A) = the EC_50_ value of a single point mutation (A)/the EC_50_ value of a single point mutation (A); AE (B) = EC_50_ (B)/EC_50_ (A); AE (A + B) = EC_50_ (A + B)/EC_50_ (A); theoretical EC_50_ (TE) of a double point mutation (A + B) = AE (A)*0.5 + AE (B)*0.5; SC = AE (A + B)/TE (A + B). According to the synergistic coefficients, the synergistic activity of the combination of different single point mutations in SDHI resistance was classified into three groups: synergic effects (SC > 1.2); additive effects (0.8 ≤ SC ≤ 1.2); and antagonistic effects (SC < 0.8).

**Table 3 microorganisms-10-00132-t003:** Correlation analysis between SDH activity and the sensitivity to SDHI in double mutations.

Pearson’s Correlations		Carboxin	Boscalid	Fluopyram	Penthiopyrad
SDH enzyme activity	Correlation coefficient (r)	−0.343	−0.468	−0.497	−0.793 *
	Significance test (P)	0.452	0.289	0.256	0.033

* significant difference.

## Data Availability

Data are available upon request.

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
