# Peer review of "Double Mutations in Succinate Dehydrogenase Are Involved in SDHI Resistance in Corynespora cassiicola"

_microorganisms, 2022, doi:10.3390/microorganisms10010132_

Round 1
Reviewer 1 Report
Citation: missing (between left margin and Abstract)
line
4 Sun Bingxue 1 , Zhu Guangxue 1 , Xie Xuewen 1 , Chai Ali 1 , Li Lei 1 , Shi Yanxia 1 *, Li Baoju 1 *
> Sun Bingxue, Zhu Guangxue, Xie Xuewen, Chai Ali, Li Lei, Shi Yanxia* and Li Baoju*
5 1 Institute of Vegetables and Flowers, Chinese Academy of Agricultural Sciences, libaojuivf@163.com
> Institute of Vegetables and Flowers, Chinese Academy of Agricultural Sciences, Beijing, China
6 * Correspondence: Li Baoju, E-mail: libaojuivf@163.com; Shi Yanxia, E-mail: shiyanxia@caas.cn.
> * Correspondence: shiyanxia@caas.cn (S.Y.); libaojuivf@163.com (L.B.)
9 Corynespora cassiicola > Corynespora cassiicola
23-24 Succinate Dehydrogenase Inhibitors; Corynespora Cassiicola; Double Mutation; SDHI-resistance; Fitness
> succinate dehydrogenase inhibitors; Corynespora cassiicola; double mutation; SDHI-resistance; fitness
29 Hevea brasiliensis > Hevea brasiliensis
32 corynespora > Corynespora
112,216 CaCl2 > CaCl2
133 sdhB/D > sdhB/D
216,362 2x,363 ml-1 > ml-1
219 equation. > equation:
243 please, increase all parts of the Figure 1. for better readability
244-245 (A, B and C) Schematic > A, B and C - schematic
256,257 sdhB > sdhB
267 2.447 > 2.472 (in Table 1)
273 1.367 > 1.357 (in Table 1)
282 C. cassiicola > C. cassiicola
308 (A-G): Comparison > A-G: comparison
308 (H): Com- > H: com-
315 2x, 316,317, 318 2x, rs > rs
317 (Car: > (carboxin:
318 Bos: > boscalid:
318 Flu: > fluopyram:
318 p = 0.21 > p = 0.021
358 C. corynespora > C. cassiicola
363 28oC > 28 °C
364 calculated.(E) > calculated. (E)
403 Table 3 > Table 3.
424 Interestingly > interestingly
433 [29]. Changes > [29] and changes
435 And previous > Previous
459 And positive > Positive
460 (Fig. 4) > (Figure 4)
464 excepted > except
478 penalty; > penalty,
482 And these > These
497 sdhB > sdhB
498 And a strong > Strong
498-499 And a strong correlation (r = -0.704, P = 0.007 Table 3) between sdhA gene expression and SDH activity was found for the sdh mutations, ...
> values for r and P are not in Table 3
> correlation between sdhA gene expression and SDH activity is not in Table 3
501-503 In addition, massive overexpression (more than 40-fold of WT) of the sdhB gene was found, but not in sdhA/C/D gene (4-5 fold of WT)(Fig. 3).
> this is not in Figure 3 (Figure 3. Spearman's correlations ...)
502 sdhB > sdhB
503 Fig. 3 > Figure 3
515-518 Author Contributions: Data curation, Bingxue Sun; Formal analysis, Bingxue Sun and Lei Li; Funding acquisition, Baoju Li; Methodology, Xuewen Xie; Resources, Guangxue Zhu and Baoju Li; Visualization, Ali Chai; Writing – original draft, Bingxue Sun; Writing – review & editing, Bingxue Sun and Yanxia Shi.
> Author Contributions: Methodology, X.X.; formal analysis, S.B. and L.L.; resources, Z.G. and L.B.; data curation, S.B.; writing – original draft preparation, S.B.; writing – review and editing, S.B. and S.Y.; visualization, C.A.; funding acquisition, L.B.
520 CARS-25), > CARS-25),
520 China(31972482), > China (31972482),
521 Program(CAAS- > Program (CAAS-
523-604 All references have to be corrected in accordance to the Instructions for Authors.
Author Response
Dear Editor and dear reviewers
Manuscript ID: microorganisms-1536717
Type of manuscript: Article
Title: Double mutations in succinate dehydrogenase are involved in SDHI resistance in Corynespora cassiicola
Authors: Bingxue Sun, Guangxue Zhu, Xuewen Xie, Ali Chai, Lei Li, Yanxia Shi
*, Baoju Li *
E-mails: sunbingxuechina@163.com, zhuguangxuer@163.cm, xiexuewen@caas.cn,
chaiali@caas.cn, caulilei@163.com, shiyanxia@caas.cn, libaojuivf@163.com
Submitted to section: Plant Microbe Interactions,
Thank you for your letter and the reviewers’ comments concerning our manuscript entitled “Double mutations in succinate dehydrogenase are involved in SDHI resistance in Corynespora cassiicola” Those comments are valuable and very helpful. We have read through comments carefully and have made corrections. Based on the instructions provided in your letter, we uploaded the file of the revised manuscript. Revisions in the text are shown using the “Track Changes”. The responses to the reviewer's comments are marked in red and presented following.
We would love to thank you for allowing us to resubmit a revised copy of the manuscript and we highly appreciate your time and consideration.
Sincerely.
Bingxue Sun
Reviewer: 1
- 4 Sun Bingxue 1 , Zhu Guangxue 1 , Xie Xuewen 1 , Chai Ali 1 , Li Lei 1 , Shi Yanxia 1 *, Li Baoju 1 *
>Sun Bingxue, Zhu Guangxue, Xie Xuewen, Chai Ali, Li Lei, Shi Yanxia* and Li Baoju*
Response: Thank you for the suggestion. This point has been revised in line 4
- 5 1 Institute of Vegetables and Flowers, Chinese Academy of Agricultural Sciences, libaojuivf@163.com
>Institute of Vegetables and Flowers, Chinese Academy of Agricultural Sciences, Beijing, China
Response: Thank you for the suggestion. This point has been revised in line 5
- 6 * Correspondence: Li Baoju, E-mail: libaojuivf@163.com; Shi Yanxia, E-mail: shiyanxia@caas.cn.
> * Correspondence: shiyanxia@caas.cn (S.Y.); libaojuivf@163.com (L.B.)
Response: Thank you for the suggestion. This point has been revised in line 7
- 9 Corynespora cassiicola > Corynespora cassiicola
Response: Thank you for the suggestion. This point has been revised in line 10
- 23-24 Succinate Dehydrogenase Inhibitors; Corynespora Cassiicola; Double Mutation; SDHI-resistance; Fitness
succinate dehydrogenase inhibitors; Corynespora cassiicola; double mutation; SDHI-resistance; fitness
Response: Thank you for the suggestion. This point has been revised in line 23-24
- 29 Hevea brasiliensis > Hevea brasiliensis
Response: Thank you for the suggestion. This point has been revised in line 29
- 32 corynespora > Corynespora
Response: Thank you for the suggestion. This point has been revised in line 32
- 112,216 CaCl2 > CaCl2
Response: Thank you for the suggestion. This point has been revised in lines 115 and 217
- 133 sdhB/D > sdhB/D
Response: Thank you for the suggestion. This point has been revised in line 136
- 216,362 2x,363 ml-1 > ml-1
Response: Thank you for the suggestion. This point has been revised in lines 218, 374 and 373
- 219 > equation:
Response: Thank you for the suggestion. This point has been revised in line 221
- 243 please, increase all parts of the Figure 1. for better readability
Response: Thank you for the suggestion. The section in Figure 1A-C has been written in "3.1 Confirmation of double mutations" in lines 225-233. The modification information is as follows, which are shown using red fonts.
To obtain C. cassiicola transformants with double mutations, site-directed mutagenesis was performed. First, the 4484 bp gene replacement vector was constructed by double-joint PCR, containing the left and right homologous arms (gray fragments in Figure 1B, 1046bp and 1167bp ) of the sdhB gene and the mutational sdhB gene (sdhB*+Trpc+neo, 2271bp, middle part in Figure 1B). Subsequently, the non-mutational sdhB gene (Figure 1A) was replaced with the gene replacement vector (Figure 1B) to generate double mutation transformants (Figure 1C). Finally, the transformants with double mutations were confirmed by following three aspects: the inserted position of the gene replacement vector, the purity of the double transformants (C. cassiicola was heterokaryotic) and the single-copy insertion of the cassette.
Transformants with double mutations connected into the locus yielded a single 2575 bp or 2791 bp band with primers P3/P4 or P5/P6, respectively, whereas those in which integration was 'random' yielded no band. These primers did not amplify any fragment from the wild-type (WT) isolate (Figure 1D-Up and 1D-Down). The purity of the transformants was confirmed by primers P7/P8 or P9/P10. Homokaryotic transformants with double mutations yielded a single 3256 bp or 2192 bp band with primers P7/P8 and P9/P10, respectively, whereas heterokaryotic transformants with double mutations yielded double bands of 3256 bp and 2091 bp by P7/P8 primers and 2192 bp and 1027 bp by P9/P10 primers. Primers P7/P8 or Primers P9/P10 amplified 2091 bp and 1027 bp from the wild-type isolate (Figure 1D-UT and 1D-TD).
We also confirmed the single-copy insertion of the cassette by Southern blotting genomic DNA digested with EcoRV. When probed with the partial sdhB gene region, the genomic DNA of the double mutation isolates digested with EcoRV had a single 4551 bp hybridized DNA fragment instead of the 3386 bp fragment found in the parental isolate (Figure 1F). Finally, the presence of each mutation was demonstrated by sequencing the entire sdhB gene.
- 244-245 (A, B and C) Schematic > A, B and C - schematic
Response: Thank you for the suggestion. This point has been revised in lines 253-254
- 256,257 sdhB > sdhB
Response: Thank you for the suggestion. This point has been revised in lines 265 and 266
- 267 447 > 2.472 (in Table 1)
Response: Thank you for the suggestion. This point has been revised in line 276
- 273 367 > 1.357 (in Table 1)
Response: Thank you for the suggestion. This point has been revised in line 282
- 282 cassiicola > C. cassiicola
Response: Thank you for the suggestion. This point has been revised in line 292
- 308 (A-G): Comparison > A-G: comparison
Response: Thank you for the suggestion. This point has been revised in line 318
- 308 (H): Com- > H: com-
Response: Thank you for the suggestion. This point has been revised in line 318
- 315 2x, 316,317, 318 2x, rs > rs
Response: Thank you for the suggestion. This point has been revised in lines 324-328
- 317 (Car: > (carboxin:
Response: Thank you for the suggestion. This point has been revised in line 327
- 318 Bos: > boscalid:
Response: Thank you for the suggestion. This point has been revised in line 328
- 318 Flu: > fluopyram:
Response: Thank you for the suggestion. This point has been revised in line 328
- 318 p = 0.21 > p = 0.021
Response: Thank you for the suggestion. This point has been revised in line 328
- 358 corynespora > C. cassiicola
Response: Thank you for the suggestion. This point has been revised in line 369
- 363 28oC > 28 °C
Response: Thank you for the suggestion. This point has been revised in line 374
- 364 (E) > calculated. (E)
Response: Thank you for the suggestion. This point has been revised in line 374
- 403 Table 3 > Table 3.
Response: Thank you for the suggestion. This point has been revised in line 414
- 424 Interestingly > interestingly
Response: Thank you for the suggestion. This point has been revised in line 437
- 433 [29]. Changes > [29] and changes
Response: Thank you for the suggestion. This point has been revised in line 446
- 435 And previous > Previous
Response: Thank you for the suggestion. This point has been revised in line 448
- 459 And positive > Positive
Response: Thank you for the suggestion. This point has been revised in line 472
- 460 (Fig. 4) > (Figure 4)
Response: Thank you for the suggestion. This point has been revised in line 473
- 464 excepted > except
Response: Thank you for the suggestion. This point has been revised in line 477
- 478 penalty; > penalty,
Response: Thank you for the suggestion. This point has been revised in line 491
- 482 And these > These
Response: Thank you for the suggestion. This point has been revised in line 495
- 497 sdhB > sdhB
Response: Thank you for the suggestion. This point has been revised in line 510
- 498 And a strong > Strong
Response: Thank you for the suggestion. This point has been revised in line 511
- 498-499 And a strong correlation (r = -0.704, P = 0.007 Table 3) between sdhA gene expression and SDH activity was found for the sdh mutations, ...
- values for r and P are not in Table 3
- correlation between sdhA gene expression and SDH activity is not in Table 3
Response: Thank you for the suggestion. The data of “strong correlation (r = -0.704, P = 0.007 Table 3)” was added in Table S5. Correlation analysis between SDH activity and expression of the sdhA gene. The revised part was shown in lines 510-512.
- 501-503 In addition, massive overexpression (more than 40-fold of WT) of the sdhB gene was found, but not in sdhA/C/D gene (4-5 fold of WT)(Fig. 3).
- this is not in Figure 3 (Figure 3. Spearman's correlations ...)
Response: Thank you for the suggestion. The data were shown in Figure 5 (Figure 5. Relative expression levels of the sdhA/B/C/D genes in the double mutation). The point has been revised in line 5156
- 502 sdhB > sdhB
Response: Thank you for the suggestion. This point has been revised in line 510
- 503 3 > Figure 3
Response: Thank you for the suggestion. This point has been revised in line 516
- 515-518 Author Contributions: Data curation, Bingxue Sun; Formal analysis, Bingxue Sun and Lei Li; Funding acquisition, Baoju Li; Methodology, Xuewen Xie; Resources, Guangxue Zhu and Baoju Li; Visualization, Ali Chai; Writing – original draft, Bingxue Sun; Writing – review & editing, Bingxue Sun and Yanxia Shi.
- Author Contributions: Methodology, X.X.; formal analysis, S.B. and L.L.; resources, Z.G. and L.B.; data curation, S.B.; writing – original draft preparation, S.B.; writing – review and editing, S.B. and S.Y.; visualization, C.A.; funding acquisition, L.B.
Response: Thank you for the suggestion. This point has been revised in lines 528-530
- 520 CARS-25), > CARS-25),
Response: Thank you for the suggestion. This point has been revised in lines 541-543
- 520 China(31972482), > China (31972482),
Response: Thank you for the suggestion. This point has been revised in lines 541-543
- 521 Program(CAAS- > Program (CAAS-
Response: Thank you for the suggestion. This point has been revised in lines 541-543
- 523-604 All references have to be corrected in accordance to the Instructions for Authors.
Response: Thank you for the suggestion. All references have been revised in accordance to Instructions for Authors in lines 546-629.

Reviewer 2 Report
Dear authors, I have read with interest the manuscript "Double mutations in succinate dehydrogenase are involved in SDHI resistance in Corynespora cassiicola".
Your work is well organized and the results are important for the future development of this field.
Author Response
Dear Editor and dear reviewers
Manuscript ID: microorganisms-1536717
Type of manuscript: Article
Title: Double mutations in succinate dehydrogenase are involved in SDHI resistance in Corynespora cassiicola
Authors: Bingxue Sun, Guangxue Zhu, Xuewen Xie, Ali Chai, Lei Li, Yanxia Shi
*, Baoju Li *
E-mails: sunbingxuechina@163.com, zhuguangxuer@163.cm, xiexuewen@caas.cn,
chaiali@caas.cn, caulilei@163.com, shiyanxia@caas.cn, libaojuivf@163.com
Submitted to section: Plant Microbe Interactions,
I would like to thank reviewer for your approval of the article. In addition, I read and corrected some grammatical errors. I hope the revised manuscript will be accepted.
Sincerely.
Bingxue Sun
Reviewer 3 Report
The paper describes a study on double mutations in succinate dehydrogenase gene of Corynespora cassiicola and their influence on resistance to succinate dehydrogenase inhibitors (SDHI). The manuscript contains some new data but it should be substantially revised before it can be considered further. Some specific comments and suggestions were given below.
Abstract is very chaotic and should be re-written. Try to organize it to tell a story of the study and not random sentences presenting some data not connected to each other. The designations introduced were not explained, so they just create confusion. Also, some sentences are hard to understand due to poor grammar and style. Please, check the MS for typos (especially species names).
Introduction is compact and sound. It gives enough background information on the subject of the study, although I miss some more justification of the importance of the study in the context of pathogen and plant disease caused. Also, describing the very origin of individual mutations could enhance the story behind using them.
Materials and methods are clearly and comprehensively described and are exhaustive, giving all the information needed to repeat the experiments. I do not have any complaints concerning this section.
Results are in general clearly shown, although with few exceptions. Figure 3 is hardly visible due to small size of the bullets. Similarly, Figure 4, especially panels C and D are overloaded with data. Please, consider changing the way of presenting these data. The description and legends, however, are fine.
Discussion and Conclusions: Second sentence (lines 411-412) misses “mutation”. In fact, the whole first paragraph is a bit chaotic and should be re-written. Try not to jump between pathogens and topics. Again, the sentence in lines 433-434 is missing something. Sentences starting with “And” are odd (lines 435-436, 459-462 and 482-483). Try to reword them.
Author Response
Dear Editor and dear reviewers
Manuscript ID: microorganisms-1536717
Type of manuscript: Article
Title: Double mutations in succinate dehydrogenase are involved in SDHI resistance in Corynespora cassiicola
Authors: Bingxue Sun, Guangxue Zhu, Xuewen Xie, Ali Chai, Lei Li, Yanxia Shi
*, Baoju Li *
E-mails: sunbingxuechina@163.com, zhuguangxuer@163.cm, xiexuewen@caas.cn,
chaiali@caas.cn, caulilei@163.com, shiyanxia@caas.cn, libaojuivf@163.com
Submitted to section: Plant Microbe Interactions,
Thank you for your letter and the reviewers’ comments concerning our manuscript entitled “Double mutations in succinate dehydrogenase are involved in SDHI resistance in Corynespora cassiicola” Those comments are valuable and very helpful. We have read through comments carefully and have made corrections. Based on the instructions provided in your letter, we uploaded the file of the revised manuscript. Revisions in the text are shown using the “Track Changes”. The responses to the reviewer's comments are marked in red and presented following.
We would love to thank you for allowing us to resubmit a revised copy of the manuscript and we highly appreciate your time and consideration.
Sincerely.
Bingxue Sun
Reviewer: 3
The paper describes a study on double mutations in succinate dehydrogenase gene of Corynespora cassiicola and their influence on resistance to succinate dehydrogenase inhibitors (SDHI). The manuscript contains some new data but it should be substantially revised before it can be considered further. Some specific comments and suggestions were given below.
- Abstract is very chaotic and should be re-written.Try to organize it to tell a story of the study and not random sentences presenting some data not connected to each other. The designations introduced were not explained, so they just create confusion. Also, some sentences are hard to understand due to poor grammar and style. Please, check the MS for typos (especially species names)
Response: Thank you for the suggestion. Abstract has been re-written in lines 8-22. The modification information is as follows.
Abstract: With the further application of succinate dehydrogenase inhibitors (SDHI), the resistance caused by double mutations in target gene is gradually becoming a serious problem, leading to a decrease of control efficacy. It is important to assess the sensitivity and fitness of double mutations to SDHI in Corynespora cassiicola and analysis the evolution of double mutations. We confirmed, by site-directed mutagenesis, that all double mutations (B-I280V+D-D95E/D-G109V/D-H105R, B-H278R+D-D95E/D-G109V, B-H278Y+D-D95E/D-G109V) conferred resistance to all SDHI and exhibited the increased resistance to at least one fungicide than single point mutation. Analyses of fitness showed that all double mutations had lower fitness than the wild type; most of double mutations suffered more fitness penalties than the corresponding single mutants. We also further found that double mutations (B-I280V+D-D95E/D-G109V/D-H105R) containing low SDHI-resistant single point mutation (B-I280V) exhibited higher resistance to SDHI and low fitness penalty than double mutations (B-H278Y+D-D95E/D-G109V) containing high SDHI-resistant single mutations (B-H278Y). Therefore, we may infer that single mutation conferring low resistance is more likely to evolve into a double mutation conferring higher resistance under the selective pressure of SDHI. Taken together, our results provide some important reference for resistance management.
- Introduction is compact and sound. It gives enough background information on the subject of the study, although I miss some more justification of the importance of the study in the context of pathogen and plant disease caused. Also, describing the very origin of individual mutations could enhance the story behind using them.
Response: Thank you for the suggestion. The origin of mutations was added in line 53-57. The modification information is as follows.
In C. cassiicola, nine genotypes that alter the amino acid sequence of the SDH complex were reported in previous studies, including B-H278R/Y, B-I280V C-S73P, C-N75S, D-S89P, D-D95E, D-H105R, and D-G109V [9-11]. Four genotypes (B-H278R/Y, C-S73P and D-G109V) were both detected in China and Japan, while some genotypes were only found in China or Japan, e.g., B-I280V, C-N75S, D-D95E and D-H105R genotypes were only found in China; C-S89P was only detected in Japan.
- Results are in general clearly shown, although with few exceptions. Figure 3 is hardly visible due to small size of the bullets. Similarly, Figure 4, especially panels C and D are overloaded with data. Please, consider changing the way of presenting these data. The description and legends, however, are fine.
Response: Thank you for the suggestion. Figures 3 and 4 have been redrawn to show the results clearly. We also upload all figures to the journal.
- Discussion and Conclusions: Second sentence (lines 411-412) misses “mutation”. In fact, the whole first paragraph is a bit chaotic and should be re-written. Try not to jump between pathogens and topics. Again, the sentence in lines 433-434 is missing something. Sentences starting with “And” are odd (lines 435-436, 459-462 and 482-483). Try to reword them
Response: Thank you for the suggestion. The whole first paragraph in Discussion has been re-written in line 421-429. These mistakes of sentences starting with “And” have been revised in line 445,447,471 and 494. The modification information is as follows.
SDHI-resistance has become a serious problem in agriculture leading to a decrease of control efficacy [20,26,27]. Several studies have shown that double mutation in sdh gene further reduces susceptibility to SDHI than single point mutation, leading to a reduction of control efficacy of SDHI fungicides [6,20,21]. However, double amino acid substitutions, resulting in SDHI-resistance, was not reported in C. cassiicola. In this research, site-directed mutagenesis was performed to create double mutations in the sdh gene, and these double mutations were analyzed for their resistance to SDHI and fitness, and further understand the evolution of resistance.
